# Postpartum Depression Prevalence and Associated Factors: An Observational Study in Saudi Arabia

**DOI:** 10.3390/medicina58111595

**Published:** 2022-11-04

**Authors:** Jamaan Alzahrani, Sameer Al-Ghamdi, Khaled Aldossari, Mansour Al-Ajmi, Dhafer Al-Ajmi, Faisal Alanazi, Abdullah Aldossary, Ahmed Alharbi

**Affiliations:** 1Department of Family & Community Medicine, College of Medicine, Prince Sattam bin Abdulaziz University, Al Kharj 11942, Saudi Arabia; 2Family Medicine Resident, King Saud Medical City, Riyadh 12746, Saudi Arabia; 3Pediatric Resident, Security Forces Hospital, Riyadh 11481, Saudi Arabia; 4General Surgery Resident, King Khalid Hospital, Al Kharj 16271, Saudi Arabia

**Keywords:** Edinburgh Postnatal Depression Scale, awareness, mother, postpartum depression, women, Saudi Arabia

## Abstract

*Background and Objectives*: Postpartum depression (PPD) is a psychological disorder in women who recently gave birth. It can present as mild to severe depression. Multiple studies across the globe have used the Edinburgh Postnatal Depression Scale to reveal the factors that affect the prevalence of PPD. Results from various published studies showed a high prevalence of PPD in Saudi Arabia. The objective of this study is to investigate the major factors that increase PPD and other associated factors. *Materials and Methods*: An observational cross-sectional study was carried out at primary healthcare centres and military hospitals in Al Kharj. The study measured the rate of PPD among Saudi women during the postpartum period. We studied 279 women aged 19–45 (mean age 31.91, SD ± 6.45) in their postpartum period and followed up to 2 months after delivery. *Results*: The prevalence of PPD in our cohort was 32.8%. Multivariate logistic regression analysis showed that previous depression diagnosis (*p* = 0.001), lower education status (*p* = 0.029), unemployment (*p* = 0.014), and delivery disposition of C-section (*p* = 0.002) remained associated with an increased risk of developing PPD. *Conclusions*: The increasing number of Saudi women with PPD demonstrates that it is a highly prevalent condition. PPD affects not only the health of the mother but also that of her baby and other family members. Our results showed that the history of diagnosed depression, lower education status, unemployment, and delivery disposition of C-sections were independent predictors of PPD development. Raising awareness and controlling the vulnerable factors could decrease the high risk of its effects.

## 1. Introduction

Postpartum depression (PPD) is a psychological health disorder in females and can present as mild to severe depression. According to DSM-5, it occurs during antenatal or the first four weeks after childbirth [1] and may last for up to a year [2]. It affects about 10–15% of all new mothers, leading to worse consequences for their entire family [3]. It also affects mother–child bonding, which can be seen as insecurity and avoidance, and the child’s growth due to the lack of breastfeeding and care [4]. A change in activity characterizes sleep patterns, appetite, and in some severe cases, suicidal thoughts or attempts [5,6]. These symptoms are comparable to major depressive disorder found in patients of the general population [7].

Worldwide, multiple studies have been conducted utilizing the Edinburgh Postnatal Depression Scale (EPDS). The EPDS scale (≥13) is indicative of PPD [8,9]. These studies have revealed many factors that affect the prevalence of PPD. These factors include domestic violence, psychiatric symptoms and family history of psychiatric illness, stressful life events, low self-esteem, lack of social support, low socioeconomic status, unplanned birth, and being left alone in labour [4,10,11]. There is an association between reduced breastfeeding and postpartum depression [12]. EPDS is valid for detecting PPD among adults [13].

The prevalence of PPD in Saudi Arabia showed variation in ten years of 14–32% from 2012–2022. Furthermore, results from published studies showed a high prevalence of PPD in Saudi Arabia [14,15,16]. Although some studies conducted and reported risk factors for PPD among Saudi women, these studies were conducted in a single governorate owing to some limitations. Some elements have not been studied in detail or discussed only one factor [17,18,19]. A recently published study conducted in Riyadh, Saudi Arabia, showed a high prevalence of PPD (38.5%) among Saudi females. However, this study included data collected during early 2018 [17]. 

PPD prevalence is different in developing and developed countries. In the latter, maternal physical and mental health are widely promoted. An article showed that a pooled prevalence of PPD was 19% in low-income and middle-income countries [20]. Recent estimates showed that the prevalence of PPD is relatively high in developing countries: Ethiopia, 23.3% [4] and Nepal, 19.4% [21]. An observational study was conducted in Dammam, Saudi Arabia, in 2014. Data were collected from 450 women who were screened for PPD using the EPDS and were interviewed. The prevalence of PPD was 17.8% [18]. In Sri Lanka, the prevalence of PPD varied at different postpartum intervals, 15.5% at ten days and 7.8% at four weeks [3]. Among Indian women, the prevalence of an EPDS score of ≥13 (which is suggestive of PPD) was 7.5% (120/1600 postpartum women) [9]. 

Today’s healthcare system faces cultural diversity and challenges, and individuals from different cultures may have different healthcare needs [22]. A large systematic review and meta-analyses of 565 studies (80 countries) estimated the global epidemiology of PPD. They found an overall prevalence of 17.22% (95% CI 16.00–18.51). They also reported that one out of five mothers experiences PPD, which is associated with high income and geographic or area development in developing countries [23]. There are considerable differences in the rates of PPD according to geographical region. A study conducted in Canada found the prevalence of PPD in Canadian mothers was 8% [24]. Likewise, a systematic review of developing (low- and lower-middle-income) countries showed the prevalence of PPD was 18.6% [25]. A recent systematic review and meta-analysis of 15 studies conducted in the Middle East reported a 27% prevalence of PPD (95% CI 0.19–0.35) [26]. Keeping in view the high prevalence of PPD in the Middle East, more specifically in Arab regions and Saudi Arabia, there is a paucity of literature on this condition [17,26,27].

Most of the studies on this topic and reported risk factors came from the western world, and we cannot relate them to Saudi culture [22,28]. Regarding maternal care, it is important to provide them with culture-appropriate facilities per their needs [29]. Lack of cultural-specific knowledge of women’s needs during and after the pregnancy hinders the change needed to improve maternal healthcare. There is a lack of studies conducted in Al Kharj, Saudi Arabia. Al Kharj is one of the main governorates of Saudi Arabia, located southeast of Riyadh. Therefore, this study was conducted to estimate the recent prevalence of PPD while also focusing on examining the factors that increase PPD and other associated factors in Al Kharj, Saudi Arabia.

## 2. Materials and Methods

The study was carried out at primary healthcare centres and military hospitals in Al Kharj, Kingdom of Saudi Arabia, with data collection occurring between December 2020 and December 2021. The study was observational and cross-sectional. The study measured the PPD rate among Saudi women during the postpartum period up to 2 months after delivery.

A validated Arabic version of the EPDS questionnaire was distributed to participants willing to participate in this survey. EPDS was used as a validated instrument to measure depressive symptoms in the postpartum period [30,31]. The questionnaires were handed to the primary health care and military hospital outpatient clinics. The trained nurses distributed the questionnaires to the target group and answered any participant questions. 

The questionnaire was divided into two parts. The first part comprised basic information and factors related to PPD. It included age, social status, education level, occupation, family financial income, number of children alive, last baby’s sex, medical history (comorbidity), history of depression, number of pregnancies, feeding pattern, whether the pregnancy was planned, and delivery methods. The second part of the questionnaire comprised a self-administered EPDS. EPDS is a reliable and valid scale to detect the symptoms of PPD in clinics and was used in multiple previous studies [9,31,32]. The validated Arabic version of EPDS with internal consistency (Cronbach’s alpha) of 0.84 was used to evaluate the status of PPD. There are a total of ten questions in EPDS. A score of >13 suggests that the person is likely to be suffering from PPD [30].

The sample size of 384 was estimated using a formula from OpenEpi (Version 3, open-source calculator—SSPropor) with confidence limits of 5% and a design effect of 1.

The data entry, wrangling, and analyses were conducted through the Statistical Package for Social Sciences (SPSS) Version 24 (IBM Corp., Armonk, NY, USA). For EPDS, a separate Excel sheet was made, and data entry and calculations were performed. The Figure was developed using GraphPad Prism Version 8.0.2 (Dotmatics, San Diego, CA, USA).

Descriptive statistics were executed in SPSS to estimate the sociodemographic and clinical characteristics of all the study participants, including the prevalence of PPD. The sociodemographic and clinical characteristics were also compared between two cohorts of mothers: women who meet EPDS’s threshold criteria for PPD and those who do not. Univariate analysis and multivariate logistic regression analysis were conducted to recognize the factors associated with the risk of developing PPD. Odds ratio (OR) with their respective 95% confidence intervals (95% CI) were reported. The level of significance was set at a *p* value of <0.05.

Ethical approval was obtained from the “Institutional Review Board of the College of Medicine at Prince Sattam bin Abdulaziz University”. The written informed consent was taken from all participants before data collection.

## 3. Results

The present study included 393 mothers with a mean age of 32.60 ± 7.51 years. Most of the study’s participants had an education at college/university/higher level (170, 43.3%). Comorbidity was observed in nearly one-fourth of the study’s participants (89, 22.6%). Thirty-nine (9.9%) patients were positive for a previous depression diagnosis. A large number of the mothers were practicing both breastfeeding and bottle-feeding (176, 44.8%). Having planned pregnancy was stated by 155 (39.4%) mothers. Normal delivery was documented for 267 participants (67.9%). The prevalence of PPD in the present study was 32.8%, according to EPDS, as shown in Figure 1. Table 1 delineates the sociodemographic and clinical characteristics of two cohorts of mothers, normal and positive for PPD, based on EPDS scoring.

As per EPDS scoring, 129 (32.8%) participants were unlikely to have PPD (EPDS score ≤8), 79 (20.1%) were possible to have PPD (EPDS score between 9 and 11), 56 (14.2%) had a fairly high possibility of depression (EPDS score of 12–13), and 129 (32.8%) were positive for PPD (EPDS score >13).

Table 2 shows the application of univariate logistic regression analysis to find the association between sociodemographic, clinical characteristics, and PPD. Of the factors found significant on univariate analysis, i.e., previous diagnosis of depression, low education status of the mothers, unemployment, delivery disposition of C-section, and mothers who combined breastfeeding with bottle-feeding, only previous depression diagnosis (OR 3.817, 95% CI 1.751–8.320, and *p* = 0.001), lower education status (OR 2.595, 95% CI 1.104–6.103, and *p* = 0.029), unemployment (OR 1.910, 95% CI 1.142–3.196, and *p* = 0.014), and delivery disposition of C-section (OR 2.342, 95% CI 1.380–3.975, and *p* = 0.002) remained associated with an increased risk of developing PPD (Table 3).

Table 3 shows the results of the multivariate logistic regression analysis. After combining all the significant factors of univariate logistic regression analysis, it was found that previous depression diagnosis (OR 3.817, 95% CI 1.751–8.320, and *p* = 0.001), lower education status (OR 2.595, 95% CI 1.104–6.103, and *p* = 0.029), unemployment (OR 1.910, 95% CI 1.142–3.196, and *p* = 0.014), and delivery disposition of C-section (OR 2.342, 95% CI 1.380–3.975, and *p* = 0.002) remained associated with an increased risk of developing PPD.

## 4. Discussion

PPD is an important public health problem that causes significant functional impairment, affects the mother and infant bonding, and causes delays in infant development. Raising awareness and controlling the vulnerable factors could decrease the high risk of its effects. However, our data showed a highly significant difference between the women who demonstrated depression versus those who did not. The depressive symptoms vary among depressive mothers. It is believed that psychosocial stressors might have an important role as a predictor of the risk of PPD. Many studies indicated that postpartum episodes began within the first postpartum month [34]. 

The prevalence of PPD in our cohort was 32.8%, which is twice as high as reported in previous studies conducted in Saudi Arabia, 17.8% in Dammam [18] and 14% in Riyadh [15]. However, these studies were conducted approximately 10 years ago. However, our findings are consistent with recent studies conducted in different governorates of Saudi Arabia. For instance, one research conducted in Jeddah reported that PPD prevalence was 21% [16]. Another study in Madinah reported 31.68% [14], and a study in Riyadh found 38.5% [17]. Nevertheless, the prevalence of PPDs is considerably high and increasing in Saudi Arabia.

Studies from another Arab country, Iran, reported a higher prevalence of PPD, 34–36% [26]. Some developing countries also showed higher values of PPD [4,20,21] compared to a few Arab countries that showed a lower prevalence of PPD. For example, in Oman, it was 10.6% [35]; in Sudan, it was 9% [32]; and in Turkey, it was 15.4% [36]. However, other non-Arab countries, such as Canada [24] and Japan [37], also showed a lower prevalence (8%) of PPD. These differences were due to different study designs, data collection techniques used in these studies, cross-cultural variation, economic conditions, social support, and were due to other country and area developments [26]. 

Education level was significantly associated with PPD; women with low education levels were more depressed as compared to others. Similarly, other studies also found that a lower education level was associated with a higher prevalence of PPD [26,38,39]. Other studies conducted in the Middle East found that low education was linked to poverty and early-age marriages, leading to PPD in young mothers. For instance, women with good educational backgrounds have better intellectual skills and coping strategies to deal with PPD [26]. In contrast, a recent study found that higher education is not a protective factor for PPD in women. They reported high PPD in nullipara with a university education [40]. The possible reason for these contrasting results is that they only include highly educated first-time mothers as the study was conducted in a university hospital, and they were slightly older. Taking a carrier break and starting parenthood is a new challenge for them, generating stress and increasing the chances of PPD. In addition, the study sample size was small, and there was a high drop-out rate, which might lead to an underestimation of results. However, a study conducted in Turkey found no significant differences between education level and PPD [36]. 

The findings show that unemployed women and women who were previously diagnosed with depression are vulnerable to PPD. This concurs with previous results showing that women growing up in socially disadvantaged environments tend to have a higher risk of PPD [41,42]. The average-income and unemployed mothers showed the highest rate of depression, while other factors had no significant effect. A study researcher reported that there are fewer chances of PPD in working women (full-time) with a professional or technical job [43]. However, in this study, most participants were highly educated and had greater awareness about their health. Nevertheless, some study results showed no significant differences in employment status, especially where the spouses support women financially [18]. It is also worth mentioning that the studies conducted in developing countries mostly have some sort of publication bias or are performed on a smaller sample size. These differences might also be due to the fact that studies either recruited mostly housewives [18] which show no association or they belong to low socioeconomic status [42], where financial stress also contributes to the development of PPD. 

A history of diagnosed depression is associated with four times more risk of having PPD than others. In line with our study results, another study reported a twenty-fold higher risk of PPD in females with previous depression (CI 95% 5, 19.72–22.42) than in those with a depression history [44]. Likewise, a study conducted in Singapore also found a strong association between PPD and a history of depression [45]. 

Mode of delivery is also associated with PPD. Surprisingly, we found that delivery through C-section had double the odds of having PPD later than others. Our findings are consistent with the recently published study by Al Nasr and Altharwi [17]. A recent systematic review of 32 studies found an increased risk of PPD after caesarean section, irrespective of any type (elective or emergency) [46]. Similarly, a Chinese study found a lower incidence of PPD in women who undergo NVD than in the C-section group [47]. Although, one study showed no association between PPD and mode of delivery [45]. Contrary to our findings, another Saudi study reported that women who delivered through normal vaginal delivery (NVD) were moderately more depressed than women who delivered through C-section [48]. The reason for the contrasting finding in this study is that the NVD group have a high percentage of multipara women, which might be the reason for the increased risk of depression in this study cohort. However, in this study, women who underwent C-sections showed a late onset of PPD (≥6 weeks), whereas our analysis does not distinguish between these timeframes. Regardless of the findings, it is attributable to the fact that women who deliver through C-section are vulnerable to many complications in the postpartum period, including restricted physical activity, risk of infection, and haemorrhage; this, in turn, increases the risk of PPD in them [17,47].

Many researchers tested the effect of stressors over a woman’s lifespan, and others have focused on the risk of pregnancy-related complications, preterm labour, and PPD [49]. Moreover, PPD is associated with a wide range of sociodemographic, financial, and lifestyle factors [23]. However, there is a need to focus on modifiable factors that, if controlled, could help alleviate PPD in most women.

The key strengths and weaknesses of this study worth mentioning are as follows. The study’s first limitation is the use of a self-administered survey form, which can be subjected to reporting bias. There is a chance of under- or over-estimating their responses as compared to a diagnosis established through a psychiatrist [3]. It also depends on the individuals’ perceptions, beliefs, and psychological health stigma level [50]. EPDS questions rely on the past seven days’ experience, which can be subjected to a recall bias. The prevalence of PPD was based on the time of assessment. Although the study is observational and cross-sectional, we followed up with the mothers for two months. While PPD may manifest up to one-year post-natal, our study’s follow-up at 12 weeks post-natal is not a weakness as it is in keeping with various studies performed in the region and because PPD has a maximal incidence within the first 12 weeks [26]. Another limitation is the use of screening tools only without comparing it with a diagnosis of PPD by a psychiatrist, as EPDS is a screening tool, not a diagnostic one. A large sample size taken from the general population of Al Kharj, Saudi Arabia, is another strength of this study. 

## 5. Conclusions

Our results showed the condition is highly prevalent among Saudi women. PPD affects not only the health of the mother but also that of her baby and other family members. History of diagnosed depression, lower education status, unemployment, and delivery disposition of C-sections were likely independent predictors of PPD development. The Edinburgh Postnatal Depression Scale is widely applicable. However, there is a need to compare findings with a clinical diagnosis of PPD. There is a need for early diagnosis of PPD, and relevant action should be taken in women with the above predictors. Future research must include a nationwide sample to re-evaluate the true prevalence and burden of PPD in Saudi Arabia. Our study included a large sample, but we suggest a larger sample size covering more areas, including additional potential factors, and using other depression scaling systems.

## Figures and Tables

**Figure 1 medicina-58-01595-f001:**
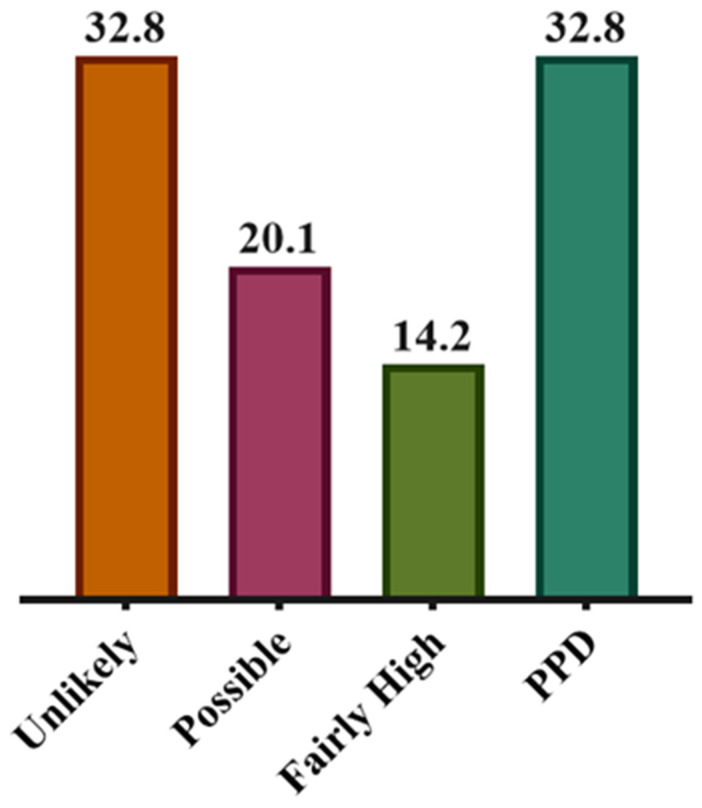
Percentage distribution of likely postpartum depression (PPD) based on EPDS.

**Table 1 medicina-58-01595-t001:** Sociodemographic and clinical characteristics of study subjects.

Characteristics	Normal *n* (%)	Positive for PPD *n* (%)
Total	264 (67.2)	129 (32.8)
Age, Years (Mean ± SD)	32.43 ± 7.42	33.00 ± 7.72
Age Distribution, Years		
<30	84 (35.3)	33 (32.7)
≥30	154 (64.7)	68 (67.3)
Comorbidity		
No	206 (79.5)	89 (71.2)
Yes	53 (20.5)	36 (28.8)
Previous Depression Diagnosis		
No	240 (94.9)	101 (79.5)
Yes	13 (5.1)	26 (20.5)
Marital Status		
Married	239 (91.6)	114 (89.8)
Divorced	18 (6.9)	9 (7.1)
Widowed	4 (1.5)	4 (3.1)
Education		
College/University/Higher	118 (44.9)	52 (40.6)
High School	92 (35.0)	34 (26.6)
Middle School	40 (15.2)	22 (17.2)
Primary School	13 (4.9)	20 (15.6)
Employment		
Employed	111 (42.7)	40 (31.3)
Unemployed	149 (57.3)	88 (68.7)
Income, SAR *		
>10,000	80 (30.8)	32 (25.2)
<5000	122 (46.9)	54 (42.5)
5000–10,000	58 (22.3)	41 (32.3)
Feeding Status		
Breastfeed	49 (20.2)	35 (28.9)
Bottle Feed	68 (28.1)	35 (28.9)
Both	125 (51.7)	51 (42.2)
Planned Pregnancy		
Yes	104 (41.8)	51 (42.5)
No	145 (58.2)	69 (57.5)
Delivery Disposition		
Normal	190 (76.9)	77 (63.1)
C-Section	57 (23.1)	45 (36.9)
Last Baby Gender		
Male	122 (49.8)	72 (58.1)
Female	123 (50.2)	52 (41.9)
No. of Pregnancies		
1–2	91 (37.0)	32 (26.9)
3–5	113 (45.9)	61 (51.3)
>5	42 (17.1)	26 (21.8)

Abbreviations: PPD, postpartum depression; SAR, Saudi riyal; C-section, Caesarean section. * Average household income in Saudi Arabia = approximately SAR 15,000 [33].

**Table 2 medicina-58-01595-t002:** Univariate regression analysis of association between sociodemographic, clinical characteristics, and PPD.

Characteristics	OR (95% CI)	*p* Value
Age Distribution, Years		
<30	Reference	
≥30	1.124 (0.686–1.841)	0.643
Comorbidity		
No	Reference	
Yes	1.572 (0.962–2.569)	0.071
Previous Depression Diagnosis		
No	Reference	
Yes	**4.752 (2.348–9.620)**	**<0.0001**
Marital Status		
Married	Reference	
Divorced	1.048 (0.457–2.406)	0.911
Widowed	2.096 (0.515–8.534)	0.301
Education		
College/University/Higher	Reference	
High School	0.839 (0.503–1.398)	0.500
Middle School	1.248 (0.675–2.306)	0.479
Primary School	**3.491 (1.615–7.545)**	**0.001**
Employment		
Employed	Reference	
Unemployed	**1.639 (1.048–2.563)**	**0.030**
Income, SAR *		
>10,000	Reference	
<5000	1.107 (0.658–1.862)	0.703
5000–10,000	1.767 (0.997–3.133)	0.051
Feeding Status		
Breastfeed	Reference	
Bottle Feed	0.721 (0.397–1.307)	0.281
Both	**0.571 (0.332–0.983)**	**0.043**
Planned Pregnancy		
Yes	Reference	
No	0.970 (0.624–1.508)	0.894
Delivery Disposition		
Normal	Reference	
C-Section	**1.948 (1.215–3.123)**	**0.006**
Last Baby Gender		
Male	Reference	
Female	0.716 (0.463–1.108)	0.134
No. of Pregnancies		
1–2	Reference	
3–5	1.535 (0.923–2.554)	0.099
>5	1.760 (0.934–3.317)	0.080

* Significant at 0.05 level. Abbreviations: PPD, postpartum depression; OR, odds ratio; CI, confidence interval; SAR, Saudi riyal; C-Section, Caesarean section. **Bold** values are statistically significant. * Average household income in Saudi Arabia = approximately SAR 15,000 [33].

**Table 3 medicina-58-01595-t003:** Multivariate regression analysis of association between sociodemographic, clinical characteristics and PPD.

Characteristics	OR (95% CI)	*p* Value
Previous Depression Diagnosis		
No	Reference	
Yes	**3.817 (1.751–8.320)**	**0.001**
Education		
College/University/Higher	Reference	
High School	0.671 (0.376–1.197)	0.177
Middle School	0.945 (0.468–1.909)	0.876
Primary School	**2.595 (1.104–6.103)**	**0.029**
Employment		
Employed	Reference	
Unemployed	**1.910 (1.142–3.196)**	**0.014**
Feeding Status		
Breastfeed	Reference	
Bottle Feed	0.768 (0.397–1.487)	0.434
Both	0.564 (0.308–1.033)	0.064
Delivery Disposition		
Normal	Reference	
C-Section	**2.342 (1.380–3.975)**	**0.002**

* Significant at 0.05 level. Abbreviations: PPD, postpartum depression; OR, odds ratio; CI, confidence interval; C-Section, Caesarean section. **Bold** values are statistically significant.

## Data Availability

The dataset supporting this article’s conclusions is the University’s property and can be made available by the authors upon request.

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
