# Peer review of "Postpartum Depression Prevalence and Associated Factors: An Observational Study in Saudi Arabia"

_medicina, 2022, doi:10.3390/medicina58111595_

Round 1

Reviewer 1 Report

Thank you for allowing me the privilege of reviewing your manuscript. Postpartum depression is indeed and important condition that is understudied in Asia, the Middle-East, and Africa. This manuscript has merit as a descriptive study on the prevalance of PPD in a city in Saudi Arabia and soundly evaluates the contributory factors. There are, however, numerous issues with proof-reeding and choice of comparisons that need to be addressed. The characterisation of factors should be improved upon as well as a clearly defined outcome i.e. EPDS of 13 and above. 

Extensive proof reading is required to allow this manuscript to be eligible for publication. In it's current format, it is quite hard to read this manuscript. Consider engaging a native speaker for this

Introduction

The introduction introduces PPD nicely and emphasises the importance of this disease entity by discussing the impacts. 

Line 61-67: 

- What is the relevance of specifying the study regarding Somali immigrants in Norway? The prevalence seems lower than in the Ethiopian and Nepalese study. Does this suggest that delivery in a developed country mitigates the risks of PPD? Immigrant predisposition to PPD may have different reasons to it compared to being a mother in a developing country.

- Is this a function of socioeconomic status or a function of ethnicity? 

- More should be done to establish the fact that there is a paucity of literature on this condition in developing countries. To that effect, is Saudi Arabia considered a developed or developing country? Would it be more reasonable to specify that there is a paucity of literature in the Middle Eastern region which has unique social, cultural, and religious norms for women?

Line 64: I would suggest stating that EPDS >= 13 is indicative of PPD in Line 50 where you first introduce it

Line 66: Specify Dammam as a city in Saudi Arabia. The international readership may not be aware of where Dammam is situated in. 

Ref 17 seems more appropriate to group with Ref 16 as it talks about a multicultural workforce. It does not discuss the fact that literature on this topic is dominated by that from Western countries

Materials and Methods

Line 81: What was the endpoint for data collection

Line 85: EPDS and not EDPS

Lines 86-87: Please state what the nature of the clinic attendances were for. PPD may occur up to a year. You have specified a cut-off of 2 months. I presume that is the typical duration of postnatal follow-up in low risk patients. In that case, this should be specified along with a mention on the standard postnatal follow-up at your institution

Line 89: The mention of the cut-off of 10 is confusing. Various studies have discussed the sensitivity and specifity trade-offs of using different cut-offs and different strata have been proposed. (https://doi.org/10.1136/bmj.m4022, https://doi.org/10.1371/journal.pone.0228666). Your analysis divides your study population into 2 cohorts using 13 as a cut-off. Beyond describing the strata, you do not use it in your univariate or multivariate analysis. Consider not discussing a cut-off of 10 to avoid confusion or placing various strata. If you must discuss the different strata and if you are keen to still describe how the population was divided into each strata, provide a citation for this. 

Line 91-92: Why is there a need to "thoroughly review the relevant literature" if you are merely describing how you structured your questionnaire.

Lines 100-103: I understand the cut-off is 13 or higher, not more than 13. I would simply discuss a cut-off of >13 being likely to be suffering of PPD (it is a screening tool, not a diagnostic one). Discussing the other strata is confusing and does not aid in describing your results.

Line 104: "A sample size of 384 was estimated using... to obtain adequate power to obtain a statistically significant result." You adjusted it to 400 but your eventual study size was 394 so this adjustment seems unhelpful to state here.

Line 110: Please cite GraphPad Prism just as your did for SPSS i.e. company, city, state, country)

Line 114-115: Again, I would just cleanly divide it at 13 and not specify too much on the other strata for PPD

Line 119: Please state the reference number for your IRB approval

Results

Line 125: WHat does comorbidity mean? You state you collected past medical history in the demographics. Does this refer to any past medical history or only pre-disposing conditions

Table 1: 

- What does Primary, Middle, and High refer to in education status? Arrange them in increasing or decreasing order. Expand it to Primary School, Middle School, and High School

- Consider specifying what the average income in Saudi Arabia is in the foot note. Alterantively, categorise it as percentiles of Saudi salaries. I presume this to mean family income and not personal income? The exact numerical significance of how many Riyals they are earning may be lost on readers

- For feeding status, place both below bootle feed and breastfeed. Does expressed breastmilk factor as "Both", "Bottlefeed" or "Breastfeed"?

- Depression Diagosis: Change this to "Previous depression diagnosis" to assure readers this refers to a prior diagnosis unrelated to the current concerns for PPD

Discussion

- Line 180-186: 

  The Qatari study uses DASS-21 and not EPDS. Is it comparable? Further, upon reading that manuscript, the difference in depression prevalence between Less than secondary level and Secondary and higher level (different strata from yours) does not reach statistical significance. I would, thus, disagree with this comparison.

  I have not been able to check each reference. It would be good to ensure all of them use a similar cut-off of 13 and above. Specifying that may reassure readers of the validity of the comparison. A quick search I did of Saudi Arabian studies shows that your PPD prevalence is comparable. 

   > https://doi.org/10.1371/journal.pone.0228666

   > https://doi.org/10.1080/17542863.2014.999691

   > 10.7759/cureus.14603

   > http://dx.doi.org/10.2147/NDT.S57556

  It may be useful to specify studies done within your country to validate taht indeed there is a higher prevalence of PPD in Saudi Arabia compared to other countries before making comparisons with other countries in the region. 

- Line 194: How does your study show that psychosocially stressed women have more depression? What does psychoscoially stressed mean? You mentioned "Depression Diagnosis" which I take to mean previous depression. Did you collect information to suggest they are actively "stressed"?

- Line 197-198: Your results show no statistical significance when considering marital status and income. Why mention these?

- Line 218: Self-administered surveys have their strengths and weakness. This is not an inherent limitation. Perhaps, expand on the type of biases that you are concerned about. You state there is a chance of under- or over-estimating their responses but how do you determine this? 

- Line 221: I would argue that the follow-up of 2 months is a weakness as PPD can manifest up to 1 year later. 

- An additional weakness is the usage a of screening tool without comparing with a diagnosis of PPD by a psychiatrist. EPDS is, still, a screening tool and not a diagnostic one.  

With the above edits, I hope to review a vastly improved manuscript in the near future. 

Reviewer 2 Report

Thank you for the opportunity to read the paper titled "Postpartum Depression Prevalence and Associated Factors: An Observational Transverse Study in Saudi Arabia". However, there are some concerned regarding this paper.

Major

1. The introduction is so short. No strong research justification and need to elaborate more.

2. The study objective is to investigate the major factors that increase PPD and other associated protective factors. In the finding were not protective factors but the factors that increased the risk of having PPD as previous depression diagnosis, lower education status, unemployment, and delivery disposition of C-section. This is contradict.

3.Prevalence and predictors of postpartum depression in Riyadh, Saudi Arabia: A cross sectional study Raneem Seif Al Nasr,.--This current paper shoud be compared from previous research done in same country as I stated. What are the differences between both. Problem statement of PPD in Saudi Arabia should be explained more, trend of prevalance of PPD in Saudi Arabia. All this important elements should be added in abstract and in the introduction.

4. Discussion section are to short need further elaboration.

5. Conclusion. Do not use point in conclusion. The predictor more appropiate in cohort study.

6. References. The authors need to check the format of the references.

For the reasons mentioned above, it is my understanding that the paper needs further improvements.

Thank you.

Round 2

Reviewer 1 Report

Dear authors,

This manuscript reads far better. It is my privilege to review this much improved version. There are a few minor edits that should be addressed in order for this manuscript to be suitable for publication in my perspective.

Inroduction:

Lines 56-58: "Additionally, women who failed to plan to breastfeed were also vulnerable to depression. There was also evidence that breastfeeding diminishes the risk of developing PPD; equally, PPD may decrease the breastfeeding rate"

- It may be more accurate to simply state that there is an association between reduced breastfeeding and postpartum depression

- I do not think that Ref 11 actually relates much to depression and breastfeeding with a main focus on domestic violence.

I appreciate and note the efforts taken to use more locally relevant data focusing on literature from Asia and the Middle-East while contrasting with Western countries. This provides a far better understanding of the background issues for the reader. Some small suggestions include;

Methods 

Line 141: Perhaps specify that the two cohorts as "between women who meet EPDS thresholds criteria for PPD and those who do not" so it is more clear who comprises the two cohorts.

Results

Line 155: I wonder if the term "mixed breast- and bottlefeeding" as opposed to "combined breastfeeding with bottle-feeding". As i understand, you have not collected information in order to specify if bottlefeeding refers to expressed breast milk or formula milk.

Lines 177-202: To save space and reduce verbosity, it may not be necessary to describe all the findings of the univariate analysis as the multivariate analysis forms the basis of your final conclusions as it corrects for various  confounding effects. Consider restructuring this section to improve flow e.g. of the factors found significant on univariate analysis i.e. previous diagnosis of depression, low education status, unemployment, C-section delivery, and breastfeeding (?versus bottle feeding and mixed breast- and bottle-feeding?) of which only a previous depression diagnosis (OR 3.8 [1.75-8.32], p 0.001), unemployment (OR 1.9 [1.14-3.20], p=0.014), and delivery by caesarean section (OR 2.34 [1.38-3.98], p=0.002) were found to be significantly associated with a meeting PPD thresholds by EPDS upon multivariate analysis.

Discussion 

Line 208-209: "...number of PPDs..." should be rephrased. Perhaps, "prevalence of PPD.."

Line 209: You state that the prevalence of PPD is rising. You cite examples to support this conclusion in the next chapter. Perhaps this sumamrising line of a high and increasing prevalence of PPD should be after the next paragraph where you describe the lower PPD prevalence in Damman and Riyadh 10 years ago but similarly high rates in recent years in other Saudi governorates.

Lines 219-226: This paragraph is good at summarising PPD prevalences in other countries. You have summarised PPD rates in other Arab, developing, and developed countries. Avoid repeating the word "some" in multiple sentences e.g. some developing, some other non-Arab. Consider rephrasing this paragraph for a better flow.

Lines 227-230: This can be placed after stating the limitation that PPD can manifest up to 1y later in Line 292. i.e. that while our study only evaluated women for PPD up to 12 weeks postnatal, it is considerably lower after that and this likely represent a timeframe of peak prevalence. 

Lines 244-248: How would you reconcile with these contrasting studies that show that higher education is not a protective factor for PPD. Could it be that there are aspects that are unclear and need further study?

Lines 257-259: Again, reconcile why these studies show a contrasting view. Are they poorly powered or was there a bias in population selection? Or are the reasons unclear and requires further study?

Lines 270-282: Once again, reconcile with these contrasting studies. It is the responsibility of the authors to use the Discussion area to explain the significance of their studies, describe differences with other studies, and suggest reasons why there are differences or what is unclear. Adjust the flow of these 2 paragraphs to state how other studies support your findings, how some do not and what the gaps are or that there may be reasons why they do not support, and cap off with a conclusion on the likely contributing factors which you have cites i.e. restricted physical activity, risk of infection, and haemorrhage

Lines 272-282: I am puzzled by the use of References 52 and 53. You use the word "likewise" for Reference 53. Likewise indicates a similarity to reference 52 or to your study. It seems to support your study findings. Reference 52, while on the surface is contrary to your study actually does state PPD is higher after 6 weeks in patients who delivery via caesarean section. Perhaps your study does not distinguish between these 2 timeframes and that overall PPD rates are higher but it follows a different pattern with later onset. 

Lines 290-292: I feel this is better rephrased as "While PPD may manifest up to one year post natal, our study's follow-up at 12 weeks post-natal is not a weakness as it is in keeping with various studies done in the region and because PPD has a maximal incidence within the first 12 weeks [Insert citation]" 

Table 1: Including the average household income of 15,000 SAR is useful. Kindly provide a reference for that. Do order the strata in "Income, SAR*" in either ascending or descending order. It would be preferable to standardise that order to each section. i.e. >10,000, 5,000-10,000, and <5,000 or in the reverse order. Do this for Table 2 as well.

Thank for you for allowing me the privilege of reviewing this revised manuscript.

Author Response

Thank you for your valuable suggestion and review for our article. We have done all the suggested changes and highlighted them in yellow in the manuscript. 

Extensive English editing has been done throughout the manuscript.

Reviewer 2 Report

Thank you for addressing my comments and suggestions. However, there are some areas that need to be improvised.

1-For title, please kindly revise to Postpartum Depression Prevalence and Associated Factors: An Observational Study in Saudi Arabia

2-Introduction needs to be rearranged based on the problem statement and justification. This section still needs to improve further. Kindly, put the prevalence of PPD in Saudi Arabia first, then compare it to other countries.

Thank you. 

Author Response

(The authors gave the same response as above.)
